# *In vitro* anti-Trypanosomatid activity and chemical profile of crude extracts of *Chromolaena hookeriana* and *Campuloclinium macrocephalum* (Asteraceae)

**Ludmila Ferreira de Almeida Fiuza¹, Ketlym da Conceição¹, Krislayne Nunes¹, Aldana Malen Corlatti²,³, Laura Cecilia Laurella²,³, Simony Carvalho Mendonça⁴, Brendo Araujo Gomes⁴, Suzana Guimarães Leitão⁴, Valeria Patricia Sülsen²,³/⁺, Maria de Nazaré Correia Soeiro¹/⁺**

¹Fundação Oswaldo Cruz-Fiocruz, Instituto Oswaldo Cruz, Laboratório de Biologia Celular, Rio de Janeiro, RJ, Brasil
²Consejo Nacional de Investigaciones Científicas y Técnicas - Universidad de Buenos Aires, Instituto de Química y Metabolismo del Fármaco, Buenos Aires, Argentina
³Universidad de Buenos Aires, Facultad de Farmacia y Bioquímica, Cátedra de Farmacognosia, Buenos Aires, Argentina
⁴Universidade Federal do Rio de Janeiro, Faculdade de Farmácia, Departamento de Produtos Naturais e Alimentos, Rio de Janeiro, RJ, Brasil

**BACKGROUND** Chagas disease (CD) and Leishmaniasis, caused by *Trypanosoma cruzi* and *Leishmania* species, are treated with outdated drugs that are toxic and have limited efficacy. New therapeutic alternatives are therefore needed. Natural products are valuable scaffolds for drug discovery, and Asteraceae species exhibit microbicidal activity.

**OBJECTIVES** This study evaluated the antiparasitic activity and selectivity of two Asteraceae species, *Chromolaena hookeriana* and *Campuloclinium macrocephalum*, against trypanosomatids.

**METHODS** *C. hookeriana* and *C. macrocephalum* extracts were profiled by ultra-high performance liquid chromatography-tandem mass spectrometry (UHPLC-MS/MS), and their cytotoxicity and antiparasitic activity were evaluated against multiple forms and strains of *T. cruzi* and *Leishmania amazonensis* (*L. amazonensis*) in 2D and 3D cultures.

**FINDINGS** UHPLC-MS/MS revealed predominantly pentacyclic triterpenoids in *C. macrocephalum*, including ursolic and oleanolic acids (negative mode) and lupeol derivatives (positive mode). *C. macrocephalum* was more effective than Benznidazole (Bz) ($EC_{50}$ = 1.17 vs 4.47 µg/mL) against epimastigotes and nine times more potent against trypomastigotes ($EC_{50}$ = 0.38 vs 3.5 µg/mL). Both extracts matched Bz activity on intracellular *T. cruzi* but exhibited time-dependent cytotoxicity. They also showed similar effectiveness to miltefosine (Mt) against *L. amazonensis* amastigotes ($EC_{50}$ = 0.36–0.70 µg/mL).

**MAIN CONCLUSIONS** The chemical profile supports the stronger antiparasitic activity of *C. macrocephalum*, likely linked to triterpenoids, and highlights Asteraceae extracts as promising candidates for drug discovery against neglected tropical diseases (NTDs).

Key words: neglected tropical diseases - *Trypanosoma cruzi* - *Leishmania amazonensis* - Asteraceae - phenotypic screening - UHPLC-MS/MS

Affecting over one billion people worldwide, neglected tropical diseases (NTDs) are a group of more than 20 diverse illnesses caused by various agents (*e.g.*, viruses, bacteria, protozoa, fungi, and ectoparasites, as well as those induced by venomous animals), closely tied to shortcomings in the public policy system for health and education, as well as deficiencies in medical infrastructure and basic sanitation systems.[1,2] In addition to their impact on healthcare systems, NTDs also affect the economies of developing countries. Despite their severity, investment from pharmaceutical companies remains low, and only a very small number of new drugs have been registered over the course of decades.[3,4,5,6]

One of these NTDs is Chagas disease (CD), also known as American Trypanosomiasis, which is caused by the hemoflagellate protozoan *Trypanosoma cruzi*.[7] Described over a century ago by Dr Carlos Chagas, CD was previously confined to rural areas of the American continent. However, due to migratory movements, these geographical boundaries have been crossed, leading to the spread of the disease to other regions. As a result, currently, there are over 6 million infected people worldwide.[8,9,10]

Different transmission pathways are associated with CD, including the classical vector-borne route, the congenital route (an important transmission pathway in both

**doi:** 10.1590/0074-02760250296

**Financial support:** FIOCRUZ, CNPq, FAPERJ, CAPES, CONICET, Universidad de Buenos Aires (UBACYT 20020220300118BA), Agencia Nacional de Promoción Cienífica y Tecnológica (PICT 2020-03061).
MNCS is researcher of CNPq and CNE.
**+ Corresponding authors:** soeiro@ioc.fiocruz.br | ⓘ https://orcid.org/0000-0003-0078-6106 / vsulsen@ffyb.uba.ar | ⓘ https://orcid.org/0000-0002-9322-5748

**Handling editor:** Adeilton Alves Brandão | ⓘ https://orcid.org/0000-0001-5877-607X

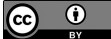

endemic and non-endemic regions), and the oral route, which is characterised by the consumption of food and/or beverages contaminated with triatomines or their faeces infected with the parasite.[11,12,13,14,15,16] Also, other important transmission routes include organ transplants and blood transfusions.[17,18,19]

In humans, CD presents itself in two clinical phases: acute and chronic. Shortly after infection, the acute phase begins, lasting up to two months, with the most evident characteristic due to patent parasitaemia that can be detected through light microscopy analysis.[20] Due to a competent vertebrate host's immune system, the parasitism is controlled but not eradicated, resulting in the second phase of the disease called the chronic stage characterised by subpatent/intermittent parasitaemia and positive serology.[19] At least one third of infected individuals will develop digestive and/or cardiac disorders that may result in death.

Regarding the treatment for CD, only two old medications, introduced in clinical therapy for more than five decades, are available: Nifurtimox (Nf) and Benznidazole (Bz). These nitroheterocyclic compounds act as prodrugs and therefore require metabolic activation by the parasite's type I nitroreductase enzyme, which reduces the nitro group to amino derivatives, generating metabolites that are highly toxic to the parasite.[21,22,23] Although effective during the acute phase, both have low efficacy during the late chronic stage. Additionally, they have other limitations such as contraindication during pregnancy, occurrence of naturally resistant strains, as well as serious side effects that frequently lead to treatment discontinuation.[7,24] For CD, the BENEFIT (Benznidazole Evaluation for Interrupting Trypanosomiasis) trial evaluated the efficacy and safety of Bz in chronic carriers, revealing its ability to significantly reduce the parasitic load but failed to prevent the progression of cardiomyopathy.[25] Two other compounds (ergosterol inhibitors - Posaconazole and the prodrug of Ravuconazole), which were also tested in clinical trials on chronic individuals, revealed lower rates of parasitological cure compared to Bz, reinforcing the urgent need to discover new drug candidates.[26,27,28]

Leishmaniasis, are a group of diseases caused by over 20 species of *Leishmania* and are transmitted via infected female sandflies (Phlebotomus and *Lutzomyia*).[29,30] Also listed within the group of NTDs, this illness is endemic in 98 countries, and causes significant economic impacts, reducing the work capacity of an infected individual.[31] *Leishmania* parasites can lead to three main clinical manifestations: visceral leishmaniasis (VL) which affects internal organs (*e.g.*, liver, spleen, and bone marrow), cutaneous leishmaniasis (CL) (with single to multiple skin ulcers, satellite lesions, or nodular lymphangitis) and mucocutaneous leishmaniasis (MCL).[30,32] More than 12 million people are infected with *Leishmania* worldwide with about 0.9 million new cases and around 30,000 deaths annually.[33]

In mammals, the *Leishmania* transmission occurs after the inoculation of metacyclic promastigotes during the bloodmeal of an infected phlebotomine. These parasites mainly invade professional phagocytic cells such as neutrophils and macrophages.[34] Inside the parasitophorous vacuole, the promastigotes transform into amastigotes which are the replicative forms present in the vertebrate hosts.[35] There are several factors that influence the disease outcome including host and parasite genetic characteristics, re-infections, as well as nutritional factors and the occurrence of comorbidities, among others.[36,37,38]

The first-line treatment for Leishmaniasis include mostly old and toxic drugs such as pentavalent antimonials that trigger numerous side effects (*e.g.*, cardiotoxicity, cirrhosis, and the induction of resistance).[39] Other therapeutic options include the use of amphotericin B as a second-line treatment, miltefosine (Mt) which has an advantage due to its oral administration, and paromomycin. However, significant limitations are associated with their use, such as prolonged treatment duration, considerable cost, teratogenicity, and induction of resistance.[40,41] Regarding new therapeutic alternatives, in recent decades, only a limited number of alternative medications or updated versions of existing ones have emerged. A Phase II study to CL evaluated the efficacy and safety of thermotherapy plus short course of Mt combination compared with thermotherapy alone. The findings showed that the combo achieves superior results than the thermotherapy alone.[42] In another study, children, and adults from Eastern Africa affected by VL were treated with two regimens of paromomycin plus Mt (14 and 28 days). The bulk of their data supports the implementation of the shorter combination regimen, which consists of 14 days of paromomycin plus Mt resulting in low toxicity events while sustaining efficacy as standard therapies.[43]

The above reported findings justify the search for new alternative treatments for both NTDs as well as a better link between pre-clinical and clinical outcomes. This is highly related to the use of more accurate methodologies including the use of 3D culture approaches. In fact, spheroids or organoids closely mimic the *in vivo* microenvironment, reproducing aspects of cell interaction and proliferation, as well as the production of extracellular matrix and cellular mediators and thus represent important experimental tools for drug discovery development.[44]

Natural products represent a continuous rich arsenal of new drug entities and have been employed since ancient times to alleviate and heal ailments.[45] Many of the drugs currently used in therapy have their origin in nature. The great chemical diversity determines their greater potential scaffold for new molecules with unique structures and potential biological activities. The analysis of new drugs approved by the Food and Drug Administration (FDA) between 1981 and 2019 revealed that 53.4% were derived from natural sources, which is particularly evident in the field of infectious diseases and cancer.[46,47]

In this context, the Asteraceae family comprises approximately 24,000 species worldwide. Asteraceae species have been studied as a potential source for novel promising drug candidates.[48] A variety of compounds have been isolated from these species, with terpenes and phenolic compounds standing out. Terpene and phenolic compounds isolated from plants belonging to this family display activity against protozoan parasites.[49] Various species of the *Eupatoriae* tribe (Asterace-

ae) have been studied, with terpene compounds such as sesquiterpene lactones (STLs) and diterpenes (DT), as well as flavonoids (FV), being the most characteristic phytochemical groups.[50,51,52]

Thus, the objective of our work was to evaluate through different *in vitro* approaches the activity of the crude extract of two species belonging to the Asteraceae family, *Chromolaena hookeriana* and *Campuloclinium macrocephalum*, against different forms and strains of *T. cruzi* and upon amastigotes of *L. amazonensis*.

Moreover, ultra-high performance liquid chromatography-tandem mass spectrometry (UHPLC-MS/MS) analysis, combined with data processing and molecular networking, was employed to obtain a qualitative characterisation of the metabolites present in the extracts, providing insights into those potentially responsible for the observed activity.

## MATERIALS AND METHODS

*Plant materials* - The aerial parts of *C. macrocephalum* (Less.) DC (BAF 802) were collected in December 2012, in Entre Rios Province and of *C. hookeriana* (Griseb.) R.M. King & H. Rob. (BAF 16123) were collected in May 2021, in Tucumán Province, both from Argentina. The plant material was identified and deposited at the Herbarium of the Faculty of Pharmacy and Biochemistry, University of Buenos Aires.

*Extraction and dilution procedures* - The crude extracts of the species *C. hookeriana* and *C. macrocephalum* were obtained by maceration with dichloromethane (10% W/V) for 5 min at room temperature, followed by filtration and evaporation using a rotary evaporator at 40ºC. The resulting crude extracts were dissolved in dimethyl sulfoxide (DMSO) to prepare stock solutions for the biological assays (concentration of 2 mg/mL) and stored at -20ºC.[52,53] Dichloromethane, a low-boiling and highly volatile solvent, is efficiently removed under these conditions. The extracts were evaporated to constant weight ensuring the absence of residual solvent.

*UHPLC-MS/MS analysis, data processing, and molecular networking* - The extracts (2 mg/mL) were analysed using a UHPLC Dionex Ultimate 3000 system (ThermoFisher Scientific, Waltham, MA, USA) coupled to an LCQ Fleet mass spectrometer (ThermoFisher Scientific). Chromatographic separation was performed using an Acquity BEH C18 column (Waters Corporation, Milford, MA, USA; 1.7 μm, 2.1 × 100 mm, 100 Å) at a flow rate of 0.45 mL.min-1, following the elution gradient: 0 min, 5% B; 5 min, 5% B; 25 min, 100% B; 30 min, 100% B; 31 min, 5% B; 36 min, 5% B. The mass spectrometer, equipped with an electrospray ionisation (ESI) source, operated in both positive and negative ionisation modes. MS spectra were acquired within the m/z range of 100-1000, using collision-induced dissociation (CID) with an energy of 35 eV. Liquid chromatography(LC)-MS/MS data were converted to the mzML format using parameters such as subset selection, msLevel 1-2, and ion polarity (positive or negative). Peak picking was performed using the vendor algorithm (msLevel 1-2) via the ProteoWizard MSConvert tool (version 3.02, Pro-

teoWizard Software Foundation, Palo Alto, CA, USA). Subsequent data processing was conducted in MZmine (version 2.53, University of Helsinki, Finland), with mass detection calibrated according to signal intensity and quality in both ionisation modes.

The ADAP Chromatogram Builder module was employed to construct chromatograms and apply wavelet-based deconvolution. Isotopic features were filtered using the Isotopic Peaks Grouper, with the most intense isotope selected as representative. Feature alignment was carried out using the Join Aligner module.

The processed results from MZmine, along with the corresponding mzML files and metadata table, were uploaded to the Global Natural Products Social Molecular Networking (GNPS) platform, utilising the feature-based molecular networking (FBMN) workflow. Standard parameters for unit-resolution data were applied in both data mining and network generation. The resulting molecular network was downloaded and further visualised and analysed in Cytoscape (version 3.9.1).

*Reference drugs solutions* - Bz and Mt were included as reference drugs in the *T. cruzi* and *L. amazonensis* assays, respectively. For *in vitro* analysis, stock solutions of reference drugs were prepared with 100% DMSO as the vehicle, all aliquots were stored at -20ºC.

*Parasites* - The epimastigotes of *T. cruzi* [Y strain, discrete typing unit (DTU) II] were obtained following the protocol described by Bombaça et al.[54] and maintained at 28ºC in liver infusion tryptose (LIT) supplemented with 10% foetal bovine serum (FBS). The parasites were collected at the exponential growth phase (five days of cell growth). Trypomastigote and intracellular forms of the Tulahuen-β gal strain (DTU VI) were obtained using L929 cell cultures (host:parasite cell ratio: 1:10).[55] For cell-derived trypomastigotes (Y and Tulahuen strains), the parasites were collected from the supernatant of previously infected L929 cell line, maintained under an atmosphere of 5% $CO_2$ at 37ºC. After 48 h of host:parasite cells interaction, the cultures were washed with phosphate buffered saline [(PBS), pH 7.4] to remove non-internalised forms and after six days the trypomastigotes collected from the supernatant. The supernatant was submitted to a centrifugation step at 4200 rpm for 10 min to retrieve the parasites in the cell pellet. To evaluate the leishmanicidal effect, *ex vivo* amastigotes from *L. amazonensis* (strain LTB0016) obtained from BALB/c male mice lesions were used throughout the study following the protocol described previously.[56]

*Mammalian cells* - L929 cell lines were used to evaluate the effect of the crude extracts on the cellular viability of mammalian host cells and to assess their activity against intracellular forms of *T. cruzi* (Tulahuen strain expressing *E. coli* β-galactosidase gene, DTU VI) using 2[57] and 3D cultures.[58] A 2D culture of rat cardiomyoblast cell lines [H9C2 (2-1)] was used to evaluate cardiotoxicity.[58] For the assessment of hepatotoxicity in a two-dimensional system, human hepatocellular carcinoma (HepG2) cells, obtained from ATCC, were cultured in Eagle's medium (MEM, GIBCO®) supplemented with 10% FBS, 100

µg/mL streptomycin, and 100 µg/mL penicillin.[59] In all assays, the cell cultures were maintained at 37ºC under an atmosphere of 5% $CO_2$. Primary mouse macrophages (PMM) were collected from Swiss male mice (18-20 g) after four days of thioglycolate (3%) inoculation PMM was collected by rinsing the animals' peritoneum with RPMI 1640, the cells were seeded at 96-wells plates and used for *in vitro* cytotoxicity assays.[56]

*Cytotoxicity in vitro tests* - To assess toxicity events L929 cells were seeded in sterile 96-well microtiter plates (flat and U bottoms as described by Fiuza et al.[58]), following the cultures were exposed or not to increasing concentrations (up to 100 µg/mL) of the extracts and reference drug. After 96 h of incubation, the cellular viability of treated cultures was compared to untreated control cells (100% of cellular viability). Cellular viability was assessed through AlamarBlue® (resazurin sodium salt) colorimetric assay comparing treated and untreated cell cultures, the results were expressed as % of cellular death to determine the $LC_{50}$ values.[57] To analyse the cardio and hepatotoxicity in 2D models, H9C2 and HepG2 cells, respectively, were seeded in a 96-well microtiter plate (Flat bottom), maintained at 37ºC. The monolayer cultures were incubated or not to increasing concentrations of the extracts and with the reference drug for 96 h. Cellular viability was assessed through PrestoBlue® fluorescence assay, and the results were expressed as a % reduction in cell viability compared to untreated control and $LC_{50}$ values determined. PMM were incubated for 48 h with increasing concentrations of the tested. To evaluate the toxicity upon PMM, the cells were incubated for 48 h with increasing concentrations of the extracts and the data expressed as % reduction in cellular viability.

*Anti-T. cruzi activity*: *epimastigote forms* - Epimastigote forms of *T. cruzi* (Y strain) were incubated or not with crescent concentrations of Bz and crude extracts. The parasite viability was assessed through PrestoBlue® and the results expressed as % of death through the comparison between treated and untreated control, to determine the $EC_{50}$ values.

*Intracellular forms in 2D system* - L929 cells (4 × 10³ cells/well) were seeded in sterile 96-well microtiter plates and infected with trypomastigotes of the Tulahuen-β-gal strain (4 × 10⁴ parasites/well) at a host cell:parasite ratio of 1:10. After the establishment of infection (48 h), increasing concentrations of the extracts were added and the cultures maintained at 37ºC for 96 h. Parasite burden was analysed by adding the substrate chlorophenol red ß-D-galactopyranoside (CPRG) followed by measurements (spectrophotometrically) at 570 nm. Bz was used as reference drug.[57,60]

*Intracellular forms in 3D system* - 3D cultures of L929 cells were obtained following the protocol described previously.[61] After 24 h, the spheroids were infected (or not, control) with trypomastigotes of *T. cruzi* (Tulahuen-β gal), using 10 parasites per host cell. Following the same protocol as the 2D system, after the establishment of infection (48 h), the spheroids were exposed

or not to crescent concentrations of Bz and the extracts. Following 96 h of drug incubation, the 3D cultures were mechanically macerated, the parasite load was measured by light microscopy quantification, and the results were expressed as % of parasite death.[58]

*Trypomastigote forms* - To analyse the compound activity against trypomastigotes of *T. cruzi* (Y strain), 10⁷/mL of parasite were added in 96-well microtiter plates and incubated for 24 h at 37ºC in RPMI culture medium in the presence or not of crescent concentrations of each extract and Bz. The death rates were assessed using a protocol adapted from Santos et al.[62] with PrestoBlue to determine the $EC_{50}$.

*Anti- Leishmania activity* - Male BALB/c mice were obtained from the animal facilities of Institute of Science and Biomodels Technology (ICTB), Fiocruz, Rio de Janeiro, Brazil, housed up to five per cage, kept in a room at 20-24ºC under a 12 h light and 12 h dark cycle, and provided with sterile water and chow ad libitum. The animals were acclimated for seven days before performing the assays. All procedures were done following Biosafety Guidelines in compliance with the Fiocruz and all animal procedures were approved by the Committee of Ethics for the Use of Animals. Animals were inoculated in the foot paws (subcutaneously) with 20 µL containing 10⁶ *Leishmania (L.) amazonensis* (MHOM/BR/77/LTB0016). Thirty days post-infection, the skin lesions were removed aseptically and mechanically dispersed by pipetting and the amastigotes were purified as reported.[62] The amastigotes (10⁷ /mL) were then exposed to increasing concentrations of the extracts and the reference compound Mt, and were incubated for 48 h at 32ºC. After drug exposure, the % of parasite death ($EC_{50}$) was determined by spectrophotometry (560-590 nm).[62]

*Statistical analysis* - The analyses were obtained through nonlinear regression analysis by GraphPad Prism v 9.0 (GraphPad Software, San Diego, USA). Statistical analysis was performed using analysis of variance (ANOVA) test with the level of significance set at $p \leq 0.05$.

*Ethics* - All animal studies were carried out in strict accordance with the guidelines established by the FIOCRUZ Committee of Ethics for the Use of Animals (CEUA L038-2017 A4).

## RESULTS

The first phenotypic analysis was performed using epimastigotes forms of *T. cruzi* (Y strain - DTU II). The parasites (10⁷/mL) were exposed for 24 h at 28ºC to increasing concentrations of the extracts and of the reference drug for CD (Bz) and the percentage of parasite death was determined by PrestoBlue reagent (560-590 nm). The data revealed that *C. hookeriana* extract showed an $EC_{50}$ value in the same range as the reference drug (6.67 and 4.47 µg/mL, respectively (Table I) while *C. macrocephalum* gave superior activity (3.7-fold) as compared to Bz, reaching an $EC_{50}$ value of 1.17 µg/mL (Table I).

As depicted in Table II, the crude extract of *C. macrocephalum* was also highly active against the intracellular forms of *T. cruzi* (Tulahuen β-gal strain - DTU

VI), reaching similar potency as Bz, with $EC_{50}$ values of 0.75 and 0.45 µg/mL, respectively. On the other hand, the *C. hookeriana* extract was 3-fold less potent than Bz with an $EC_{50}$ value of 1.52 µg/mL (Table II). Simultaneously, cytotoxicity experiments were conducted with fibroblasts from the L929 cell line, revealing $LC_{50}$ values ranging from 0.8 - 1.2 µg/mL, while Bz showed a $LC_{50}$ of > 13 µg/mL (Table II).

The L929 spheroids were infected (ratio = 20 parasites: 1 host cell) with the Tulahuen strain and exposed to increasing concentrations of the crude extracts and the reference drug. After 96 h of incubation, the spheroids were macerated and quantified by light microscopy using a Neubauer chamber to determine the $EC_{50}$. In parallel, toxicity assays were also conducted in the 3D model (L929 cells). Regarding the antiparasitic activity of Bz and of both crude extracts, similar results were achieved when 2D and organoid L929 cell lines were compared, reaching $EC_{50}$ values ranging from 0.74 to 1.02 µg/mL (Table III). Regarding mammalian host cell toxicity, both extracts affect viability, with *C. macrocephalum* being the less toxic towards the spheroid cultures with a $LC_{50}$ = 8.3 µg/mL and selectivity index (SI = 8. The reference drug (Bz) did not exert a toxicity profile up to the maximum tested concentration (Table III).

Next, parasites obtained from the supernatant of previously infected L929 cell cultures exposed or not to increasing concentrations of crude extracts. Treatment with Bz was conducted in parallel. After 24 h of treatment, the parasite death analysis was performed using PrestoBlue (PB). Also, cytotoxicity was further investigated in two additional cell lineages: (i) cardiac cells - H9C2 and (ii) hepatic cells - HepG2. Both extracts proved to be more active than Bz against trypomastigotes, with *C. macrocephalum* standing out the higher potency, presenting an $EC_{50}$ of 0.38 µg/mL after 24 h of incubation, being approximately 9-fold more active than the reference drug (Table IV). Regarding toxicity assays, while Bz did not result in toxic events up to the maximum tested concentration, both extracts displayed higher toxicity profile on the two distinct cell types resulting SIs of 12 and 34 on H9C2 and HepG2 cells, respectively (Table IV).

The assays on different forms of *T. cruzi* from various strains (DTU II and VI) provided a dataset that motivated us to investigate the activity of the extracts against another protozoan from the same family, which causes CL. Further, the *C. macrocephalum* and *C. hookeriana* extracts were evaluated through *L. amazonensis*, at the same time, we also assessed toxicity on mammalian cells using peritoneal mouse macrophages, which represent the preferred host cells for these parasites. The data demonstrated that both crude extracts exhibited potent leishmanicidal activity, with $EC_{50}$ values similarly to the reference drug (Mt), ranging from 0.36 to 0.7 µg/mL (Table V). However, concerning toxicity, Mt did not show toxic effects up to the maximum tested concentration, while both extracts showed $LC_{50}$ values of 3.9 and 6.7 µg/mL, respectively (Table V).

TABLE I

Activity against epimastigote forms of *Trypanosoma cruzi* after 24 h of incubation with *Campuloclinium macrocephalum*, *Chromolaena hookeriana* extracts and Benznidazole (Bz) at 28°C

| Reference drug/extracts | $EC_{50}$ - µg/mL |
|---|---|
| Bz | 4.475 (2.855 to 7.014) |
| *C. macrocephalum* | 1.176 (0.605 to 2.283) |
| *C. hookeriana* | 6.674 (4.148 to 10.74) |

TABLE II

Activity of *Campuloclinium macrocephalum*, *Chromolaena hookeriana* crude extracts and Benznidazole (Bz) against intracellular forms of *Trypanosoma cruzi* (Tulahuen strain), cytotoxicity in L929 cells, and their corresponding selectivity index (SI)

| Reference drug/extracts | $EC_{50}$ - µg/mL | $LC_{50}$ - µg/mL | SI |
|---|---|---|---|
| Bz | 0.45 (0.29 to 0.68) | > 13 | > 28 |
| *C. macrocephalum* | 0.75 (0.40 to 1.39) | 0.805 (0.39 to 1.63) | 1 |
| *C. hookeriana* | 1.52 (0.92 to 2.49) | 1.139 (0.66 to 1.94) | ≈1 |

TABLE III

Evaluation of anti-*Trypanosoma cruzi* (intracellular forms of Tulahuen strain) activity ($EC_{50}$) in 3D matrices of infected L929 cells, toxicity ($LC_{50}$), and selectivity indexes (SIs) using *Campuloclinium macrocephalum*, *Chromolaena hookeriana* crude extracts and to Benznidazole (Bz) after 96 h of incubation

| Reference drug/extracts | $EC_{50}$ - µg/mL | $LC_{50}$ - µg/mL | SI |
|---|---|---|---|
| Bz | 0.74 (0.405 to 1.363) | > 26 | > 35 |
| *C. macrocephalum* | 1.02 (0.503 to 2.067) | 8.30 (5.199 to 13.26) | 8 |
| *C. hookeriana* | 1.01 (0.426 to 2.409) | 2.96 (1.874 to 4.681) | 3 |

TABLE IV

Evaluation of anti-*Trypanosoma cruzi* (cultured derived trypomastigotes of Tulahuen strain) activity ($EC_{50}$), toxicity ($LC_{50}$) profile using different cell lineages (H9C2 and HepG2) and corresponding and selectivity indexes (SIs) after 24 h of incubation

| Reference drug/extracts | $EC_{50}$ - µg/mL (Range) | H9C2 Cells | | HepG2 Cells | |
|---|---|---|---|---|---|
| | | $LC_{50}$ - µg/mL (Range) | SI | $LC_{50}$ - µg/mL (Range) | SI |
| Bz | 3.589 (2.127 to 6.057) | > 26 | > 7.2 | > 13 | > 4 |
| *C. macrocephalum* | 0.385 (0.215 to 0.688) | 4.695 (2.658 to 8.290) | 12 | 13.03 (7.107 to 23.90) | 34 |
| *C. hookeriana* | 1.393 (0.804 to 2.412) | 3.402 (2.060 to 5.617) | 2 | 6.086 (3.092 to 11.98) | 4 |

Bz: Benznidazole.

TABLE V

Analysis of anti-*Leishmania* activity ($EC_{50}$) and toxicity profile in peritoneal macrophages (MØ) after 48 h of exposure to the extracts using *Campuloclinium macrocephalum*, *Chromolaena hookeriana* extracts and miltefosine (Mt), and their corresponding selectivity indexes (SIs)

| Reference drug/extracts | $EC_{50}$ - µg/mL | $LC_{50}$ - µg/mL | SI |
|---|---|---|---|
| Mt | 0.361 (0.176 to 0.739) | > 40 | > 111 |
| *C. macrocephalum* | 0.433 (0.200 to 0.936) | 6.72 (1.320 to 34.21) | 15.6 |
| *C. hookeriana* | 0.706 (0.353 to 1.414) | 3.90 (1.765 to 8.614) | 5.6 |

UHPLC-MS/MS analysis, combined with data processing and molecular networking, was carried out to obtain a qualitative characterisation of the metabolites present in the extracts, which can be related to the observed anti-*T. cruzi* and anti-*Leishmania* activities. This integrative approach provides valuable insights into the relationship between metabolite composition and bioactivity.

Liquid chromatography-electrospray ionisation (LC-ESI) (+/-)-MS/MS were evaluated in both negative and positive ionisation modes, in conjunction with molecular networking, to qualitatively profile the triterpenoids present in *C. hookeriana* and *C. macrocephalum* extracts. Our analysis prioritised the differentiation between triterpenoid acids, alcohols, and their functional derivatives.

Negative ionisation mode proved especially effective for detecting triterpenoid acids such as ursolic acid and oleanolic acid, both with the molecular formula $C_{30}H_{48}O_3$, and presenting deprotonated molecular ions $[M-H]^-$ at $m/z \approx 455$ (Table VI). These compounds exhibited characteristic MS/MS fragmentation patterns involving sequential neutral losses of $CO_2$ (44 Da), $H_2O$ (18 Da), $CH_4$ (16 Da), and CO (28 Da), consistent with decarboxylation processes and ring cleavages within the A and E rings.[63] Additional fragment ions at m/z 409, 379, 321, 293, and 275 were detected and interpreted as products of allylic rearrangements and retro-Diels-Alder (RDA) reactions, contributing to the discrimination between ursane and oleanane skeletons.[64] These fragment ions were annotated in Molecular Network 1 - Fig. 1, which was constructed from MS/MS data, and were predominantly observed in the *C. macrocephalum* extract. Notably, this extract also exhibited the most promising results in the antiparasitic assays,

suggesting a possible correlation between the presence of these triterpenoid acids and the observed bioactivity.

In contrast, positive ionisation mode favoured the detection of triterpenoid alcohols, such as lupeol (m/z 427.9) and 3-O-acetyl-lupeol (m/z 469.9), along with several unidentified derivatives (Table VII). These compounds typically formed stable $[M+H]^+$ ions and underwent fragmentation through neutral losses of water (18 Da), formic acid (46 Da), and acetic acid (60 Da), in addition to retro-Diels-Alder (RDA)-type cleavages within the polycyclic core. The molecular network derived from this analysis - Fig. 2 showed a predominance of these ions in the *C. macrocephalum* extract, reinforcing the consistency observed between ionisation modes and the species-specific chemical profiles.[64]

The LC-MS/MS chromatograms of *C. macrocephalum* and *C. hookeriana* obtained under both positive and negative ionisation modes are shown in Fig. 3. The combination of negative and positive ionisation data enabled a more comprehensive profiling of pentacyclic triterpenoids in the plant extracts, encompassing both acidic and alcoholic forms, as well as their oxygenated and acetylated derivatives. The diversity of annotated ions, including m/z 455, 409, 379, 427.9, and 469.9, reflects the rich chemical complexity of these Asteraceae species and suggests that such compounds may act as bioactive metabolites or serve as chemotaxonomic markers.

This integrative approach further underscores the potential of mass spectrometry coupled with molecular networking as a robust strategy for exploring metabolic diversity in complex botanical matrices, especially when dealing with structurally related isomers that are challenging to resolve using conventional methods.

TABLE VI

Annotated ursolic/oleanolic acid derivatives identified via liquid chromatography-tandem mass spectrometry (LC-MS/MS) in molecular network 1

| Compounds | Rt (min) | Molecular formula | [M-H]⁻ *m/z* | [M+H]⁺ *m/z* | MS/MS | Proposed compound |
|---|---|---|---|---|---|---|
| 1 | 10.56 | - | 451.0 | - | 433, 405, 387, 375, 289, 245 | NI |
| 2 | 11.63 | $C_{30}H_{48}O_3$ | 455.0 | - | 425, 409, 379, 321, 293, 275 | ursolic acid |
| 3 | 12.22 | $C_{30}H_{48}O_3$ | 454.8 | - | 437, 409, 391, 387, 379, 293, 275 | oleanolic acid |
| 4 | 12.89 | - | 512.9 | - | 495, 467, 453, 407, 389, 327, 319, 309, 291, 275 | NI |
| 5 | 13.0 | - | 433.0 | - | 414, 389, 387, 375, 345, 289, 245 | NI |
| 6 | 13.38 | - | 512.9 | - | 495, 475, 467, 453, 437, 407, 379, 355, 333, 323, 309, 291, 279 | NI |
| 7 | 13.47 | $C_{32}H_{50}O_4$ | 497.0 | - | 451, 437, 391, 363, 349, 329, 305, 293, 275, 259, 247, 231 | 3-*O*-acetyloleanolic acid |
| 8 | 14.12 | - | 437.0 | - | 419, 391, 373, 361, 343, 325, 315, 275, 247, 231 | NI |
| 9 | 14.69 | - | 439.0 | - | 425, 409, 393, 375, 363, 349, 331, 293, 277, 269, 259, 245 | NI |
| 10 | 15.0 | - | 480.9 | - | 463, 449, 435, 421, 375, 365, 353, 321, 291, 245 | NI |
| 11 | 16.2 | - | 480.9 | - | 463, 435, 421, 375, 357, 331, 313, 277 | NI |

Rt: retention time; NI: not identified.

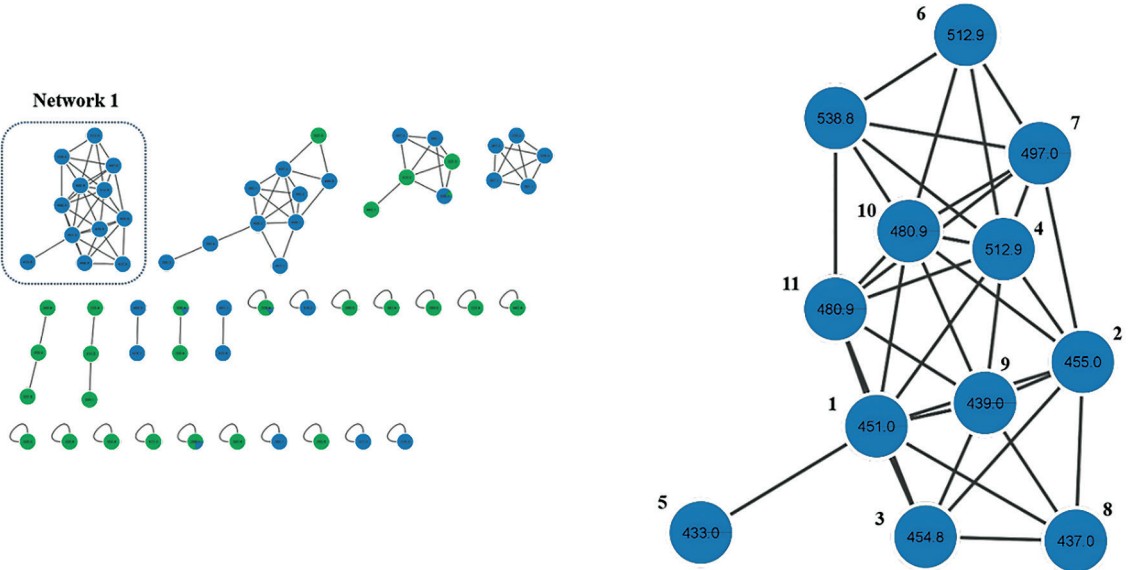

Fig. 1: molecular networking based on liquid chromatography-tandem mass spectrometry (LC-MS/MS) data in negative ionisation mode for *Chromolaena hookeriana* (green) and *Campuloclinium macrocephalum* (blue) extracts. Focus on network 1, highlighting the most abundant ions found in the most active extract (*C. macrocephalum*).

Taken together, the integration of biological assays with UHPLC-MS/MS and molecular networking highlights the antiparasitic potential of *C. macrocephalum* and *C. hookeriana*, while providing a framework for future studies aimed at isolating and validating the active metabolites responsible for the observed effects.

## DISCUSSION

Natural products are a relevant source of new chemical entities in drug discovery and development related to several diseases including those of infectious origin.

Their extensive diversity and intricate structural compositions provide a large array of new scaffolds of pharmaceutical interest, such as for the therapy of NTDs.[56-61] As promising biological properties of natural products such as the Asteraceae species have been reported,[65,66,67,68] our aim was to investigate the potential activity of dichloromethane extracts from the aerial parts of *C. hookeriana* and *C. macrocephalum* against *T. cruzi* and *L. amazonensis* parasites following a well-standardised protocol of phenotypic analysis for novel drug candidates for CD and CL, respectively.[55,56,60] Regarding

TABLE VII

Annotated lupeol derivatives identified via liquid chromatography-tandem mass spectrometry (LC-MS/MS)
in molecular network 1

| Compounds | Rt (min) | Molecular formula | [M-H]⁻ *m/z* | [M+H]⁺ *m/z* | MS/MS | Proposed compound |
|---|---|---|---|---|---|---|
| 12 | 12.69 | $C_{32}H_{52}O_2$ | - | 469.9 | 453, 435, 417, 393, 355, 337, 319, 277, 259, 251 | 3-*O*-acetyl-lupeol |
| 13 | 12.25 | $C_{30}H_{50}O$ | - | 427.9 | 411, 393, 313, 295, 277, 259, 209 | lupeol |
| 14 | 14.55 | - | - | 451.9 | 435, 417, 399, 373, 337, 319, 293, 275, 259, 241 | NI |
| 15 | 12.63 | - | - | 465.9 | 449, 431, 389, 351, 333, 315, 291, 273, 255 | NI |
| 16 | 11.60 | - | - | 441.9 | 425, 407, 389, 343, 327, 309, 291, 267, 249 | NI |
| 17 | 14.83 | - | - | 453.9 | 437, 419, 395, 375, 359, 339, 321, 295, 261 | NI |

Rt: retention time; NI: not identified.

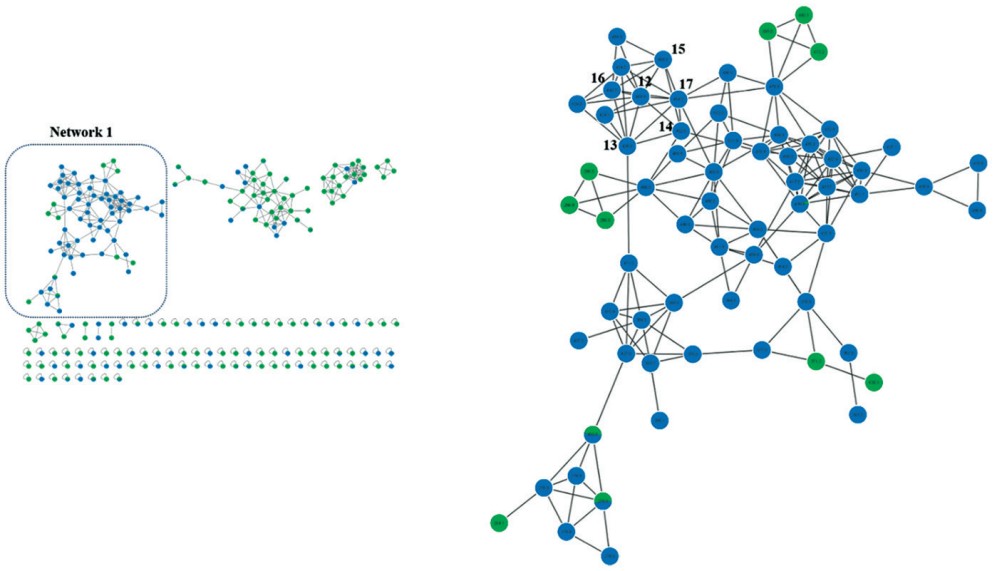

Fig. 2: molecular networking based on liquid chromatography-tandem mass spectrometry (LC-MS/MS) data in positive ionisation mode for *Chromolaena hookeriana* (blue) and *Campuloclinium macrocephalum* (green) extracts. Focus on network 1, highlighting the most abundant ions found in the most active extract (*C. macrocephalum*) and lupeol derivatives.

the activity exhibited upon epimastigotes our data are aligned and corroborate previous studies that demonstrated the promising activity of Asteraceae family extracts against *T. cruzi*, with inhibitory concentration values lower than 2 µg/mL[69] corroborating our present data with *C. macrocephalum*.

The promising data justified further *in vitro* studies of the crude extracts against the relevant forms for vertebrate hosts besides exploring the toxic profile against mammalian host cells.[57] Over intracellular forms both extracts showed a trypanocidal activity against *T. cruzi*, highlighting *C. hookeriana* that exhibited similar potency as Bz (EC$_{50}$ values of 0.75 and 0.45 µg/mL, respectively), corroborating previous studies with Aster-

aceae family, as *Mikania periplocifolia*.[67] The extract of *C. macrocephalum* revealed a moderate toxicity profile as reported using crude extracts of Asteraceae.[70]

There are several gaps between the outcomes from *in vitro* and *in vivo* studies, and one is related to the use only of 2D systems that are less reproducible as an organoid matrix.[71] Thus, further assays were presently conducted using different three-dimensional models following well-established protocols.[58] Both crude extracts showed similar activity as Bz giving EC$_{50}$ values ≈ 1 µg/mL and 0.74 µg/mL, respectively, when organoids were used as host cells. As reported by Fiuza et al.[58] spheroids were less susceptible as compared to 2D cultures when incubated with the extracts while Bz

did not exert toxicity until the maximum tested concentration. About toxicity evaluation, a more toxic profile was detected when H9C2 were assayed, suggesting a potential cardiotoxic effect of both extracts.

We next evaluated their effect on another form of the parasite relevant to mammalian infection: trypomastigote forms (Y strain - DTU II). Both extracts were more effective than Bz, corroborating the data from Laurella et al.[67] using Asteraceae fractions.

The assays on different forms of *T. cruzi* from various strains (DTU II and VI) provided a dataset that motivated us to investigate the activity of the extracts against another protozoan from the same family, which causes CL. Both crude extracts were evaluated on amastigote forms of *L. amazonensis*, purified from cutaneous mice lesions.[62] Our findings revealed a potent leishmanicidal activity of *C. macrocephalum* and *C. hookeriana*, with $EC_{50}$ values comparable to those of Mt (0.36 to 0.7 µg/mL). The efficacy demonstrated by the crude extracts against intracellular forms of *L. amazonensis* corroborates Fróes et al.[68] findings, which also revealed leishmanicidal activity of *Vernonanthura brasiliana* extract against promastigotes.

Beyond the antiparasitic activity, chemical profiling was conducted to gain insight into the potential bioactive compounds involved. In this context, species from the Asteraceae family, such as *C. hookeriana* and *C. macrocephalum*, both members of the *Eupatorium* complex,

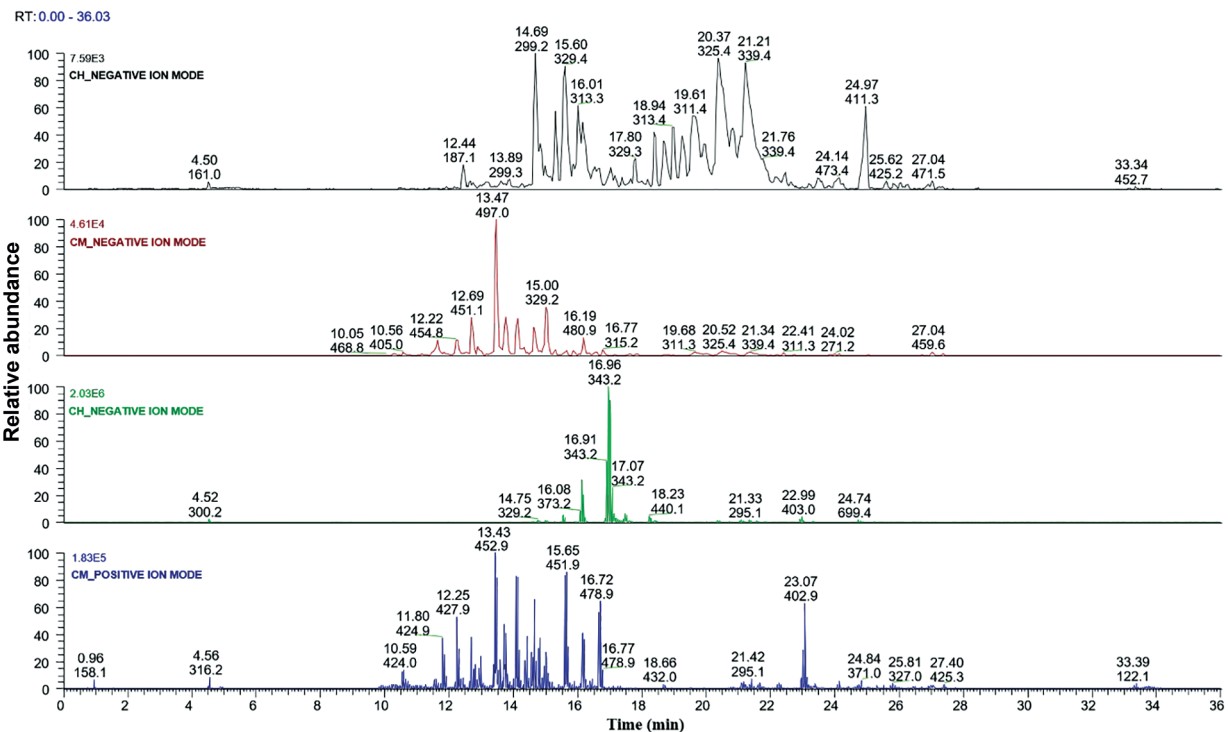

Fig. 3: comparative liquid chromatography-tandem mass spectrometry (LC-MS/MS) profiles of *Chromolaena hookeriana* and *Campuloclinium macrocephalum*, in both ionisation modes. For each peak, the retention time (Rt) and the corresponding m/z values are displayed in the chromatograms.

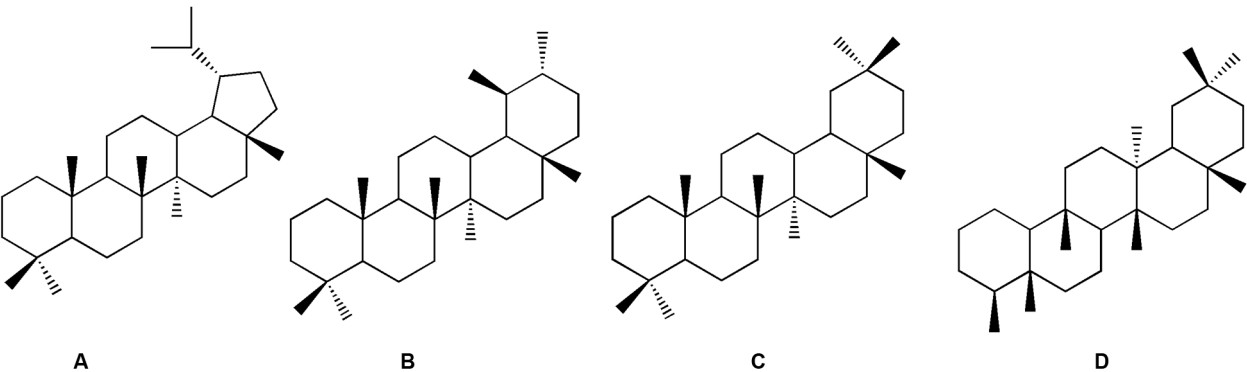

Fig. 4: basic skeletons of pentacyclic triterpenoids: lupane (A), ursane (B), oleanane (C) and friedelane (D) types. Adapted from Wei et al.[74]

are known to produce a wide array of secondary metabolites, particularly pentacyclic triterpenoids.[72,73] These compounds are typically found as complex mixtures of structural isomers bearing distinct core skeletons (*e.g.*, lupane, ursane, oleanane, and friedelane) (Fig. 4), which poses a considerable challenge for their annotation by mass spectrometry due to high spectral similarity and frequent co-elution in plant extracts.[63]

*In conclusion* - Our present findings demonstrate the antiparasitic effects of two Asteraceae species, *C. macrocephalum* and *C. hookeriana* that justify additional studies. Then, considering the promising antiparasitic activity of *C. macrocephalum* ($EC_{50} \leq 1$ µg/mL and SI $\geq$ 8), fractionation, isolation and purification of its active compounds are currently underway, followed by additional phenotypic evaluations. In parallel, UHPLC-MS/MS analysis combined with molecular networking has provided a qualitative overview of the metabolites present, offering valuable insights into potential chemical contributors to the observed activity. Overall, these results highlight the potential of natural products as a source of novel drug candidates for NTDs, for which safe and effective therapies remain urgently needed.

## ACKNOWLEDGEMENTS

To Nadia T Mirakian and Rachel Napoles Rodriguez from Instituto de Química y Metabolismo del Fármaco, CONICET - Universidad de Buenos Aires, Buenos Aires 1113, Argentina for their technician support.

## AUTHORS' CONTRIBUTION

MNCS and VPS - funding acquisition, conceptualisation, project administration, resources, data curation, supervision; LFAF and MNCS - formal analysis and investigation; LFAF, KC, KN, AMC, LCL and MNCS - methodology; MNCS - validation; LFAF, LCL, VPS, MNCS, SCM, BAG and SGL - writing-original draft and writing-review & editing; SCM, BAG and SGL UHPLC-MS/MS analysis. This investigation is part of the activities within the "Research Network Natural Products against Neglected Diseases" (ResNet NPND) (http://www.resnetnpnd.org/). The authors declare no conflict of interest. All authors have read and agreed to the published version of the manuscript.

## DATA AVAILABILITY

The data supporting the findings of this study are available within the article. Additional data are available from the corresponding author upon reasonable request.

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

# OPEN PEER REVIEW

Memórias do IOC thanks the anonymous reviewers for their contribution to the peer review of this work.

## FIRST REVIEW ROUND

REVIEWERS' COMMENTS

### REVIEWER #1

There is an adequacy for all the points.

### REVIEWER #2

My comments:
Some points and suggestions are presented below for adjustment:
Line 165- add "and" to indicate that both species are from the same country;
Line 171- Inform the maceration time (days/hours).
Chloroform extracts are not suitable for biological evaluation. How was the absence of traces of chloroform verified in the crude extract?
Verify the correct orthography of the botanical family name Asteraceae (lines 459, 474…).
Figure 3- Specify in the legend that the retention time (RT) and the m/z values are shown for each peak in the chromatograms."
Considering the article's multidisciplinary coverage, I suggest including the structural representation of a pentacyclic terpenoid.

Other aspects:
a) Adequacy of the abstract - Good;
b) Originality and importance of the contribution for the development of the field of study - The current treatment options used for Chagas disease and leishmaniasis relies on outdated drugs, whose efficacy is often associated with significant adverse effects. These limitations, together with the fact that both diseases are classified as neglected tropical diseases, underscore the urgent need for novel therapeutic compounds that offer greater effectiveness and improved safety. In this context, research dedicated to the prospecting and development of new substances continues to be highly valued.
c) Methodology, results and discussion - Good;
d) References - Good;
e) Figures and tables - Good.

AUTHORS' RESPONSE TO THE REVIEWERS

Dear Editor,
Memórias do Instituto Oswaldo Cruz
Rio de Janeiro, 9th  February 2026

Dear Editor,
Please find enclosed the revised manuscript entitled "*In vitro* evaluation of crude extracts of *Chromolaena hookeriana* and *Campuloclinium macrocephalum* upon Trypanosomatid parasites" by Fiuza et al., 2026, submitted for your consideration for publication in the Memórias do Instituto Oswaldo Cruz. This manuscript is original, has not been published, and is not under consideration for publication elsewhere. All authors have reviewed and approved the final version of the manuscript and are in full agreement with the journal's current guidelines and conditions. The current treatment of Chagas disease and Leishmaniasis basically remains on old drugs that have limited activity and display side effects. These limitations, together with the fact that both belong to tropical neglected diseases' group highlights the urgent need to find more effective and safer alternative compounds. In this context, the search for new drugs is urgently needed, and thus our study aims to provide an important contribution in this field meriting the publication. Also, all relevant contributions for the referees were incorporated into the revised MS as well as the response letters that follow below.

Thank you very much.
Yours sincerely,
Dr. Maria de Nazaré C. Soeiro
Laboratório de Biologia Celular
Instituto Oswaldo Cruz, Fundação Oswaldo Cruz, Av. Brasil 4365 - CEP 21040-360, Rio de Janeiro – Brasil
Email: soeiro@ioc.fiocruz.br , Tel. +55 21 25621368

Responses to Referee's comments:
Reviewer 1: There is an adequacy for all the points.
Answer: Many thanks for the time and positive evaluation of our manuscript.

Reviewer 2 comments:
Answer: Many thanks for the time and positive evaluation of our manuscript.

Some points and suggestions are presented below for adjustment:
Line 165- add "and" to indicate that both species are from the same country.
Answer: Thanks, the sentence was revised as suggested.

2. Line 171- Inform the maceration time (days/hours).
Answer: Thanks, the sentence was revised as suggested and the time included.

3. Chloroform extracts are not suitable for biological evaluation. How was the absence of traces of chloroform verified in the crude extract?
Answer: Thanks for the comment. In fact, we have used dichloromethane as solvent for extraction. Then, the extracts were taken to complete dryness under reduced pressure using a rotary evaporator at temperatures below 40 °C. Dichloromethane, a low-boiling and highly volatile solvent, is efficiently removed under these conditions. The extracts were evaporated to constant weight ensuring the absence of residual solvent. Also, the toxicity of the extracts was inspected by using different mammalian cells cultures as depicted in Methodology section (Cytotoxicity in vitro tests) and reported in Tables II-V. To avoid misunderstanding, the sentence was revised in the methodology section.

4. Verify the correct orthography of the botanical family name Asteraceae (lines 459, 474…).
Answer: Thanks, it was fully revised.

5. Figure 3- Specify in the legend that the retention time (RT) and the m/z values are shown for each peak in the chromatograms."
Answer: Thanks, it was fully revised and included in the Figure legend.

6.Considering the article's multidisciplinary coverage, I suggest including the structural representation of a pentacyclic terpenoid.
Answer: Many thanks. It was incorporated as Figure 4.

Other aspects:
a) Adequacy of the abstract - Good;
b) Originality and importance of the contribution for the development of the field of study - The current treatment options used for Chagas disease and leishmaniasis relies on outdated drugs, whose efficacy is often associated with significant adverse effects. These limitations, together with the fact that both diseases are classified as neglected tropical diseases, underscore the urgent need for novel therapeutic compounds that offer greater effectiveness and improved safety. In this context, research dedicated to the prospecting and development of new substances continues to be highly valued.
c) Methodology, results and discussion - Good;
d) References - Good;
e) Figures and tables - Good.
Answer: Many thanks for improving our revised MS.

## SECOND REVIEW ROUND

REVIEWERS' COMMENTS

### REVIEWER #1

No other comments.

### REVIEWER #2

The suggested recommendations have been duly incorporated into the revised version of the manuscript. The implemented modifications are appropriate and contribute to improving the clarity and comprehension of the manuscript.

