## [Reviewer Report · FIRST REVIEW ROUND - REVIEWERS COMMENTS]

## REVIEWER #1

There is an adequacy for all the points.

## REVIEWER #2

My comments:

Some points and suggestions are presented below for adjustment:

Line 165- add “and” to indicate that both species are from the same country;

Line 171- Inform the maceration time (days/hours).

Chloroform extracts are not suitable for biological evaluation. How was the absence of traces of chloroform verified in the crude extract?

Verify the correct orthography of the botanical family name Asteraceae (lines 459, 474…).

Figure 3- Specify in the legend that the retention time (RT) and the m/z values are shown for each peak in the chromatograms.”

Considering the article’s multidisciplinary coverage, I suggest including the structural representation of a pentacyclic terpenoid.

Other aspects:

a) Adequacy of the abstract - Good;

b) Originality and importance of the contribution for the development of the field of study - The current treatment options used for Chagas disease and leishmaniasis relies on outdated drugs, whose efficacy is often associated with significant adverse effects. These limitations, together with the fact that both diseases are classified as neglected tropical diseases, underscore the urgent need for novel therapeutic compounds that offer greater effectiveness and improved safety. In this context, research dedicated to the prospecting and development of new substances continues to be highly valued.

c) Methodology, results and discussion - Good;

d) References - Good;

e) Figures and tables - Good.

## AUTHORS’ RESPONSE TO THE REVIEWERS

Dear Editor,

Memórias do Instituto Oswaldo Cruz

Rio de Janeiro, 9th February 2026

Dear Editor,

Please find enclosed the revised manuscript entitled “*In vitro* evaluation of crude extracts of *Chromolaena hookeriana* and *Campuloclinium macrocephalum* upon Trypanosomatid parasites” by Fiuza et al., 2026, submitted for your consideration for publication in the Memórias do Instituto Oswaldo Cruz. This manuscript is original, has not been published, and is not under consideration for publication elsewhere. All authors have reviewed and approved the final version of the manuscript and are in full agreement with the journal’s current guidelines and conditions. The current treatment of Chagas disease and Leishmaniasis basically remains on old drugs that have limited activity and display side effects. These limitations, together with the fact that both belong to tropical neglected diseases’ group highlights the urgent need to find more effective and safer alternative compounds. In this context, the search for new drugs is urgently needed, and thus our study aims to provide an important contribution in this field meriting the publication. Also, all relevant contributions for the referees were incorporated into the revised MS as well as the response letters that follow below. Thank you very much.

Yours sincerely,

Dr. Maria de Nazaré C. Soeiro

Laboratório de Biologia Celular

Instituto Oswaldo Cruz, Fundação Oswaldo Cruz, Av. Brasil 4365 - CEP 21040-360, Rio de Janeiro – Brasil

Email: soeiro@ioc.fiocruz.br , Tel. +55 21 25621368

Responses to Referee’s comments:

Reviewer 1: There is an adequacy for all the points.

Answer: Many thanks for the time and positive evaluation of our manuscript.

Reviewer 2 comments:

Answer: Many thanks for the time and positive evaluation of our manuscript.

Some points and suggestions are presented below for adjustment:

Line 165- add “and” to indicate that both species are from the same country.

Answer: Thanks, the sentence was revised as suggested.

2. Line 171- Inform the maceration time (days/hours).

Answer: Thanks, the sentence was revised as suggested and the time included.

3. Chloroform extracts are not suitable for biological evaluation. How was the absence of traces of chloroform verified in the crude extract?

Answer: Thanks for the comment. In fact, we have used dichloromethane as solvent for extraction. Then, the extracts were taken to complete dryness under reduced pressure using a rotary evaporator at temperatures below 40 °C. Dichloromethane, a low-boiling and highly volatile solvent, is efficiently removed under these conditions. The extracts were evaporated to constant weight ensuring the absence of residual solvent. Also, the toxicity of the extracts was inspected by using different mammalian cells cultures as depicted in Methodology section (Cytotoxicity *in vitro* tests) and reported in Tables II-V. To avoid misunderstanding, the sentence was revised in the methodology section.

4. Verify the correct orthography of the botanical family name Asteraceae (lines 459, 474…).

Answer: Thanks, it was fully revised.

5. Figure 3- Specify in the legend that the retention time (RT) and the m/z values are shown for each peak in the chromatograms.”

Answer: Thanks, it was fully revised and included in the Figure legend.

6.Considering the article’s multidisciplinary coverage, I suggest including the structural representation of a pentacyclic terpenoid.

Answer: Many thanks. It was incorporated as Figure 4.

Other aspects:

a) Adequacy of the abstract - Good;

b) Originality and importance of the contribution for the development of the field of study - The current treatment options used for Chagas disease and leishmaniasis relies on outdated drugs, whose efficacy is often associated with significant adverse effects. These limitations, together with the fact that both diseases are classified as neglected tropical diseases, underscore the urgent need for novel therapeutic compounds that offer greater effectiveness and improved safety. In this context, research dedicated to the prospecting and development of new substances continues to be highly valued.

c) Methodology, results and discussion - Good;

d) References - Good;

e) Figures and tables - Good.

Answer: Many thanks for improving our revised MS.

---

## [Reviewer Report · REVIEWERS COMMENTS]

## REVIEWER #1

No other comments.

## REVIEWER #2

The suggested recommendations have been duly incorporated into the revised version of the manuscript. The implemented modifications are appropriate and contribute to improving the clarity and comprehension of the manuscript.